# Influence of Printing Parameters on Self-Cleaning Properties of 3D Printed Polymeric Fabrics

**DOI:** 10.3390/polym14153128

**Published:** 2022-07-31

**Authors:** Ayat Adnan Atwah, Mohammed Dukhi Almutairi, Feiyang He, Muhammad A. Khan

**Affiliations:** 1College of Designs and Arts, Umm Al-Qura University, Al Taif Road, P.O. Box 715, Mecca 21955, Saudi Arabia; 2School of Aerospace, Transport, and Manufacturing, Cranfield University, Cranfield MK43 0AL, UK; m.almutairi@cranfield.ac.uk (M.D.A.); feiyang.he@cranfield.ac.uk (F.H.); 3Centre for Life-Cycle Engineering and Management, Cranfield University, College Road, Cranfield MK43 0AL, UK

**Keywords:** 3D printing, self-cleaning fabric, microscopic features, printing parameters, (TPU 98A), thermoplastic elastomers (TPE felaflex), thermoplastic co-polyester (TPC flex45)

## Abstract

The processes for making self-cleaning textile fabrics have been extensively discussed in the literature. However, the exploration of the potential for self-cleaning by controlling the fabrication parameters of the fabric at the microscopic level has not been addressed. The current evolution in 3D printing technology provides an opportunity to control parameters during fabric manufacturing and generate self-cleaning features at the woven structural level. Fabrication of 3D printed textile fabrics using the low-cost fused filament fabrication (FFF) technique has been achieved. Printing parameters such as orientation angle, layer height, and extruder width were used to control self-cleaning microscopic features in the printed fabrics. Self-cleaning features such as surface roughness, wettability contact angle, and porosity were analyzed for different values of printing parameters. The combination of three printing parameters was adjusted to provide the best self-cleaning textile fabric surface: layer height (LH) (0.15, 0.13, 0.10 mm) and extruder width (EW) (0.5, 0.4, 0.3 mm) along with two different angular printing orientations (O) (45° and 90°). Three different thermoplastic flexible filaments printing materials were used: thermoplastic polyurethane (TPU 98A), thermoplastic elastomers (TPE felaflex), and thermoplastic co-polyester (TPC flex45). Self-cleaning properties were quantified using a pre-set defined criterion. The optimization of printing parameters was modeled to achieve the best self-cleaning features for the printed specimens.

## 1. Introduction

Textiles are considered one of the most used materials in daily life. These materials are known for their breathability, softness, and low cost of raw materials. They are widely used in the clothing industry, furnishings, and industrial fabrics [1]. The textile industry is one of the fastest-growing and has been subject to innovative improvements. Functional characteristics of textile surfaces, such as self-cleaning, antimicrobial, anti-sticking, and waterproofing features are of increasing interest in both scientific and industrial sectors [2]. Self-cleaning provides a lot of benefits in various industries [3]. This property was inspired by natural phenomena that can be noticed on the leaves of lotus plants, rice plants, or animals, such as butterfly wings and fish scales [4]. A self-cleaning surface is potentially beneficial for many commercial products for economic, aesthetic, and environmental reasons [5,6]. Overall, it can maintain a clean and pollution-free surface either by decomposing adsorbed stains and chemicals or preventing dust and other pollutants from being set on the surface [7].

In recent years, an effort has been made to introduce innovations in the fabrication of textile materials and the growth of conventional textiles. The development of self-cleaning textile fabrics using finishing methods is one of the most promising research areas in textile technology [8]. It has enormous potential for product improvement in the clothing and health industries. This technology supports the maintenance of a pollution-free environment and effectively decreases cleaning efforts by saving time, laundry cost, and preserving a considerable amount of water and energy [3,9]. Self-cleaning fabric surfaces have previously been produced by creating micro-nano rough structures on surfaces with low surface energy or altering low-surface-energy materials to rough surfaces [10]. Many methods have been developed for the preparation of hydrophobic and photolytic coatings, such as dip-coating (sol-gel), spin coating, spray coating, electro-spraying, chemical etching, chemical vapor deposition (CVD), hydrothermal treatment (HTT), electrospinning, layer-by-layer self-assembly (LBL) and wet chemical synthesis. However, preparation using these methods requires different chemicals and processes. Some of them are affordable and easy to handle, but others are complicated, expensive, and suitable only at a laboratory scale. Several chemical solutions have been used along with nanotechnology to enhance coatings for desirable surface roughness or lower surface energy. A nano-layer of anatase TiO_2_ and other nanoparticles have been successfully applied to the textile fabric’s surface or incorporated into the fiber itself [11].

Identifying the key structural features in textile fabric self-cleaning surfaces is essential for improving the liquid-resistant/repellent surface properties of the fabric. Critical microscopic features such as surface roughness, porosity, and wettability of a textile fabric are the primary influences on self-cleaning properties at the surface [12,13,14,15,16,17,18,19]. However, there is no record of exploration of the potential for introducing features of self-cleaning by controlling fabrication parameters at the microscopic level. Control of fabrication parameters is not easy using conventional manufacturing techniques. For this reason, surface-coating methods are needed to introduce self-cleaning features for most textile fabrics.

The current state of 3D printing technology provides an opportunity to control fabrication parameters [20,21] during fabric manufacturing so that desirable properties [22,23,24,25,26] such as self-cleaning features at the woven structural level [27,28,29,30,31] may be introduced. Fabrication of 3D printed textile fabrics are using a low-cost fused filament fabrication (FFF) technique has been carried out. Printing parameters such as orientation angle, layer height and extruder width were used to control self-cleaning microscopic features in the printed fabrics. Self-cleaning features such surface roughness, wettability contact angle, and porosity were analyzed for different values of printing parameters. The combination of three printing parameters was adjusted to provide the best self-cleaning textile fabric surface: layer height (LH) (0.15, 0.13, 0.10 mm) and extruder width (EW) (0.5, 0.4, 0.3 mm) along with two different angular printing orientations (O) (45° and 90°). Three different thermoplastic-flexible-filaments printing materials are used: thermoplastic polyurethane (TPU 98A), thermoplastic elastomers (TPE felaflex), and thermoplastic co-polyester (TPC flex45). The self-cleaning features were quantified using a pre-set criterion. The optimization of printing parameters was modelled to identify the optimum self-cleaning properties for the printed specimens. The devised method developed and model of optimization can be used to estimate the self-cleaning properties of printed fabric if the inputs of the printing parameters are known.

## 2. Theory and Hypothesis

### 2.1. Definition of Printing Parameters

This study focuses on three printing parameters (layer height (LH), extruder width (EW), and raster orientation angle (O)) and the possibility of developing a 3D printed self-cleaning textile fabric. It identifies the significance of the fabric’s microscopic features, such as porosity, surface roughness, and wettability, along with the aesthetic look after optimizing these features. These microscopic features were obtained and controlled by using different sets of printing parameters, as shown in Table 1.

Further, the influence of these features on mechanical strength at the fabric woven-structure level was tested, along with their effect on self-cleaning ability. The fundamental definitions of printing parameters, the microscopic features, and their interrelationship are provided below to comprehend this influence and self-cleaning ability.

Layer height (LH) mm: The height of each layer in the printed specimen. It influences the number of layers printed for a given thickness of specimen, as shown in Figure 1.A fixed number of layers with different values of layer height will provide different values of fabric thickness.Extruder width (EW) mm: The width of the printed material is dependent on the nozzle size of the selected printer. The influence of extruder width on self-cleaning features can only be evaluated if the overall fabric dimensions remain constant, as shown in Figure 1 [32].Orientation angle (O)°: The path on which the nozzle moves on each layer of the (FFF) 3D printer part as in Figure 2.

The mentioned printing parameters were adjusted in the 3D printer software in this research. Other parameters such as nozzle temperature (°C), print speed (mm/s), and infill density (%) were selected based on the recommendation of the printer OEM (i.e., original equipment manufacturer). These parameters mainly depend on hardware capacity and quality. In the presented research, printing parameters including (LH), (EW), and (O) are analyzed to develop fabrics with self-cleaning ability. Parameters, mainly dependent on hardware, are maintained on the values advised by the printer OEM manual.

### 2.2. Definition of Microscopic Features

Commonly, as mentioned in previous studies, the self-cleaning ability can be attributed to material concerning many microscopic features such as porosity (%), surface roughness (μm), and wettability contact angle (°C). Surface roughness is a factor of surface texture. Any surface is made of three elements, roughness, waviness, and form with the wavelength of the surface particles [33]. It refers to the structural surface part, ranging from smooth to rough, depending on the material. These structures consist of a pattern and the main direction. A roughness value can be calculated on a profile (line) or a surface (area). The profile roughness parameter (Ra, Rq) is more common, as shown in Figure 3 [34,35]. The surface roughness can affect the water repellence for hydrophobic materials [36].

Wettability is described as the ability of a liquid to wet a surface; it is determined with the help of contact angle measurement. The three-phase contact point of solid–liquid-air interface is defined as the contact angle [37]. It is one of the most influential parameters during water droplet impact on a solid substrate [16,38] as shown in Figure 4. The wetting behaviour of a solid surface is controlled by both surface roughness and the geometric structure of the surface.

The porosity was defined as the ratio of void space within the boundaries of solid material to the total volume [39]. The structure of a textile contains pores between the fibers and the yarns. Pore dimension and distribution are a function of the fabric geometry [13], as shown in Figure 5. These features are mainly responsible for this ability. They are being checked and optimized based on the experimental changed value for the chosen printing parameters.

### 2.3. Interrelationship of Printing Parameters and Microscopic Features

New 3D printing methods have extended the capability of 3D printers and offer a new area of usage. Many efforts have explored the possibility of fabricating textiles using a 3D printer. Different textures can be printed by controlling the nozzle’s height and the amount of material extrusion [40]. Surface roughness, porosity, and wettability are the critical features of a self-cleaning surface. The input printing parameters can control these three features if FDM-based 3D printing is used to fabricate textile fabrics. The present study uses different combinations of printing parameters and examines their relationship with the obtained features. If a surface can attain a required surface roughness and a possible porosity level, its wettability can be affected by these two parameters. Once wettability reaches a certain angle, the surfaces can be considered superhydrophobic or hydrophobic. So, the basic principle for any self-cleaning surface requires suitable surface roughness to generate an acceptable contact angle for a water droplet to float on the surface instead of being absorbed. The 3D printing of textile fabrics of different materials provides the control on the values of possible roughness and porosity levels with ease compared to the conventional weaving process. The roughness can be changed due to the changes in the printing parameters, such as increasing the layer height can affect the surface texture, leading to increase printing time and lower surface roughness. Similarly, the porosity changes as the void spaces get affected by increasing the layer height or extruder width, but it can be minimized. As the surface roughness was affected, the surface’s wettability was also affected. Printing orientation also affects surface roughness as the measuring files showed differences in the roughness values. Changes in layer height have also provided highly variable results considering materials rather than surface properties. These parameters jointly decide how much heat is put in per unit of time and hence to what extent the melted material can settle [30].

## 3. Methodology

### 3.1. Design of Experiment

The experiments were mainly divided into three parts:The specimens were designed and printed with three different parameters and three different materials. The chosen values provided an excellent printing quality and the other parameters remained constant for all the samples. The prepared samples were tested based on an experimental scheme to evaluate the self-cleaning ability.The microscopic features (porosity-roughness and wettability), which are mainly responsible for this ability, were measured and recorded to evaluate and compare the best values for self-cleaning between the three chosen materials.The data were analyzed to define the optimal self-cleaning number.

The experimental outputs were used in analytical calculations to find the relationship between changes in printing parameters and microscopic features. The detailed experimental scheme is shown in Figure 6.

### 3.2. FFF Materials

The structures were fabricated by using a (Raise3D pro-2) printer with different types of filaments for (FFF)-based such as flexible filaments (Thermoplastic Polyurethane (TPU 98A), Thermoplastic elastomer (TPE fila flex), Thermoplastic Co-Polyester (TPC flex 45), PA12 NYLON, and Polyethylene terephthalate Glycol-Modified (PETG). The mentioned filament materials in Table 2 were selected for their fabric-like properties, such as flexibility and softness, and hence can be a suitable replacement for conventional textile fabric material.

### 3.3. Samples Preparation

Two layers were printed in two orientation angles of 45° and 90°. The cross-section of the filament is a flat rectangle measuring 6 × 10 cm printed under different sets of parameters as shown in Figure 7. The Autodesk Inventor 2020 CAD software was used to design the specimen in STL file format and imported to the printer Idea Maker 4.2.0 software. Idea Maker 4.2.0 software was used to set a series of printing parameters. Most parameters were recommended on the default set values during the printing process, apart from the selected parameters shown in Table 3.

The final samples for a self-cleaning textile were set and tested through specific measurement as in Figure 8.

### 3.4. Experimental Setup and Measurements Procedure

#### 3.4.1. The Measurement Method for Porosity

The main advantage of using digital image analysis to assess the porosity of the textile specimen is that there is greater accuracy and higher duplicability. Digital images of part of the specimens were taken by Dino-lite digital microscope. As the images taken were coloured, we transferred them to grayscale. The images of the fabrics with known magnifications were imported into a computer and converted into binary images. Then the porous area and the total area of the fabric were extracted. They were calculated by MATLAB software, as shown in Figure 9.

#### 3.4.2. The Measurement Method of Surface Roughness

In this study, the measurement was evaluated by the white-light interferometer. The surface roughness definition presented as the average of the standard deviation, which shows that a higher peak means higher roughness. If the peaks are higher, the water droplet will be moving on the surface and will not go inside the pores. Physical sample-surface-roughness parameters (Ra and Rq) were first obtained. The surface roughness Ra was measured quantitatively in μm from the filtered profiles as shown in Figure 10 and Figure 11.

#### 3.4.3. The Measurement Method of Wettability

The tools and techniques used to measure the contact angle of liquid drops with solid surfaces are also being developed from very basic to some advanced techniques. Direct Goniometric Method: The direct goniometric method is the most used method for measuring liquid contact angles on solid surfaces. The instrument consists of four essential components: (1) a horizontal surface to mount a liquid or solid sample; (2) a micropipette to form a liquid droplet on the surface; (3) an illuminating source; and (4) a telescope assembled with a protractor eyepiece as shown in Figure 12. The angle measured through the droplet at the interference of the three phases, i.e., solid, liquid and vapour, is referred to as the water contact angle (WCA) [17].

#### 3.4.4. The Unit Cell Measure

The unit cell for the three filament TPU 98A—TPE fila flex—TPC 45 flex were measured under a Dino-lite digital microscope with 50× magnification as shown in Figure 12. The unit cell measurements are provided in Table 4 as in Figure 13.

#### 3.4.5. Data Process

The porosity, surface roughness and wettability values were recorded and evaluated for the three different materials. A multiple linear regression (MLR) model was used to show the relationship between the printing parameters changes and the mentioned microscopic features. Therefore, their effect on the self-cleaning ability is shown in the equation.
(1)$$y=b_{0}+b_{1}x_{1}+b_{2}x_{2}+b_{3}x_{3}\;$$
where *y*ˆ is the microscopic feature (porosity, roughness, and wettability), and *x_1_* represents the layer height, *x_2_* represents the extruder width, *x_3_* represents the orientation angle, and *b_0−3_* is the estimated regression coefficient that quantifies the association between the parameters X and the dependent variable *y*ˆ.

The printing parameters are converted from the original values in Table 1 into standardized dimensionless values to eliminate the effects of differences in properties, such as dimension and order of scale between different variables, thus making the effect sizes of different variables comparable as shown in Table 5.

The most basic principle of self-cleaning ability is forming a spherical water droplet that can remove the dirt particles by affecting the surface properties [41]. To explain the optimal self-cleaning number, we applied wettability as the most influential parameter during water droplet impact on a textile fabric. It was used to justify the optimal self-cleaning number. If the contact angle >100°, the solid surface is referred to as a hydrophobic surface: the water rolls off and the surface has the minimum self-cleaning ability; if it is less than <90°, the surface becomes hydrophilic in nature which means the angle is = 0° and the water will be absorbed by the surface. If the contact angle approaches >150°, the surface is termed a superhydrophobic surface and it has the best self-cleaning ability [42].

The self-cleaning number was evaluated based on the linear equation.
(2)$$\mathrm{Tan}\;\left(\theta \right)=\frac{100}{60}\;\theta ={\;\tan}^{-1}(\frac{10}{60})\;\left(Y-Y_{1}\right)=\frac{\left(y_{2}-y_{1}\right)}{\left(X_{2}-X_{1}\right)}\left(X-X_{1}\right)$$

As Y = is the self-cleaning no, (*Y_1_* = 0), (*Y_2_* = 100), (*X_1_* = 90) and (*X_2_* =150).

Self-cleaning number = 1.6 (*X* − 90)

As *X* = wettability
(3)Self−cleaning number=1.6 (wettability – 90)

## 4. Result and Discussion

### 4.1. The Influence of Printing Parameters on the Microscopic Features

The influence of the chosen printing parameters (layer height, extruder width, and orientation angle) on the microscopic features were evaluated by plotting them separately with respect to affecting self-cleaning ability.

#### 4.1.1. The Influence of Printing Parameters on Porosity

This study investigated the layer height effects for the three materials (TPU 98A, TPE filaflex, TPC 45 flex). Layer height is the difference between one layer and the next deposited layer. We hypothesized that the size of gaps (porosity) would change if the printing parameters changed. The result showed the changing of gap sizes (porosity) according to the layer height. Increasing the layer height can increase the porosity by about 5% to 10% for the three materials. The (TPU 98A) obtained porosity values between 54% and 65%, about a 10% increase for one layer height. The porosity number is almost the same regarding layer height as the range changed slightly when the layer height increased. The average value remained the same. For (TPC 45flex), it showed some variation in the layer height in porosity by about 10% to 15% at a higher layer height and a lower layer height, respectively. The porosity increased slightly when the layer height changed from 0.10 mm to 0.15 mm. The 0.15 mm layer height had the highest number range of 66%, and the 0.10 mm layer height had the lowest.

The layer affected the porosity number in lower values, but when it went higher, the values became constant. Comparison between layer height influence (TPE) porosity values to the (TPU and TPC) showed a significant difference in porosity numbers by approximately 40% or more. It showed porosity values obtained between 9% and 18% for the chosen layer height. It slightly changed porosity by about 2% to 5% at higher and lower layers, respectively. Therefore, it can be found that the changes in porosity numbers were dependent on the material itself rather than on the layer height changes. The density (g/m^3^) for the (TPU and TPC) was 1.14, 1.16 g/m^3^ respectively, and the (TPE) was 1.09 g/m^3^. The results are represented in the diagram shown in Figure 14.

For the three different materials (TPU 98A, TPE filaflex, TPC 45flex), the extruder-width-influencing porosity number remained constant when it increased. For (TPU 98A), the values started from 54% to 64%, from the lowest extruder width value to the highest extruder width, about 10%. Similarly, for the (TPE) the extruder width did not change the porosity number for the same extruder width. When the extruder width increased, the range remained the same at 10%. Increasing the extruder width did not affect the porosity number, but it changed other parameters. The (TPE) porosity values were the lowest among the other materials. They was a significant change in the porosity number of almost 40%. The change we had from higher extruder width to lower extruder width was about 7%. It meant that the value did not change drastically for the same material. Individually, layer height and extruder width did not impact the porosity number, but when they were combined, they could affect the porosity numbers, as shown in Figure 15.

Orientation had no impact on the porosity numbers for the three materials. The porosity numbers’ change ranged from lower values to higher values but remained constant. For (TPU), the porosity number for 45 and 90 orientations had the same range, increasing from 58% to 65% and 59% to 66%, respectively. When the orientation changed, the range remained similar, about 7%. However, the (TPE) porosity values were the lowest of the other materials. They showed a considerable change in porosity of almost 40% in 45 and 90 orientation. As for this material, we had a change of about 10% in respect of orientation, as shown in Figure 16. For example, if the total porosity of the material shifted from 50% to 55% it would create an impact on surface roughness which will also affect the wettability .

#### 4.1.2. The Parameters’ Influence on Surface Roughness

The differences in the surface roughness behavior profiles of printed fabric at different layers of height for the three different materials (TPU 98A, TPE filaflex, TPC 45flex) showed a variation of values as the range changed when the layer height increased. It is common in textiles to use Mean Absolute Deviation (MAD), calculated below along with Ra. It is the most valuable and standard parameter for analyzing the surface structure [43]. We hypothesized that printing inputs such as line width and layer height have an apparent direct influence on roughness: there are more comprehensive lines and thicker layers; there are changes in print temperature; speed also affects roughness; and the print orientation angle can affect how the final part looks and feels [30]. Changing the print layer height and extruder width produced samples that spanned relatively RQ ranges between (0.04–0.65 μm), yet showed significant differences in the wetting ability. For (TPU 98A), increasing the layer height can increase the surface roughness. It was shown that 0.10 mm of layer height had the lowest surface roughness (0.06 μm), whereas the maximum roughness occured at 0.15 mm of layer height (0.45 μm). The increase, which was affected by the other parameters, is considerable for a one layer height. The average amount of the increase remained the same. For (TPC 45flex), the result showed that the surface roughness of 0.10 mm of layer height had the lowest value of the three materials (0.04 μm), and maximum roughness occurred at 0.15 mm of layer height (0.38 μm). However, the (TPE) showed the highest roughness values of 0.15 mm of layer height (0.65 μm), whereas it decreased at 0.10 mm of layer height (0.16 μm). Therefore, the surface of the fabric with the highest layer height had the highest roughness values for the three materials, as the surface became rougher. It can be seen that high roughness values decrease when the layer height is lower. The results are represented in the diagram shown in Figure 17.

For extruder width, the (TPU 98A) shows that 0.30 mm of extruder width had the lowest surface roughness (0.06 μm), whereas the maximum roughness occurred at 0.50 mm of extruder width (0.48 μm). For (TPC 45flex), 0.30 mm of extruder width had the lowest surface roughness (0.04 μm), and the maximum roughness occurred at 0.50 mm of extruder width (0.53 μm). However, the (TPE) showed the highest roughness values of 0.50 mm of extruder width (0.65 μm), and the roughness occurred at 0.30 mm of extruder width (0.16 μm). It indicated that the increase in roughness was affected by extruder width. The increase was considerable for one extruder width, which has been affected by the other parameters. The average value of the increase remains the same. Therefore, the surface with the highest width has the highest roughness values for the three materials. The results are represented in the diagram in Figure 18.

For the same material there was a significant change with the two different orientation angles (45–90). The surface roughness profile at a 45° measuring direction with narrower peaks to valleys distribution increased in the measuring direction angle of 90°. Between the 45° and 90° measuring direction, the surface roughness change remained relatively constant, with a minor variation of approximately (0.06–0.45) μm. The three materials (TPU 98A), (TPC 45flex), and (TPE) showed the highest roughness values in a printing orientation of 90°, and the values decreased in a printing orientation of 45°, as shown in Figure 19. One possible way to justify the differences in the effect of the various parameters would be the changes in the material density that caused the highest roughness between the three chosen materials, which was approximately 0.65 μm. For changes in individual print parameters, our results offered acceptance margins in respect of producing a specific desired surface roughness. It suggests that manufacturers, whether private users or companies, have considerable room for adjusting printing parameters within the current range that affects the physically perceived surface roughness.

#### 4.1.3. The Parameters’ Influence on Wettability

The connection between surface roughness and wettability indicates that a higher surface roughness improves the water repellence for hydrophobic materials, and the water contact angle is dependent on it. The angle measured through the droplet at the interference of the three phases, i.e., solid, liquid, and vapor, is referred to as the water contact angle (WCA). If the contact angle > 90° the solid surface is referred to as a hydrophobic surface and the water droplet will roll off the surface, and if it is < 90° the surface becomes hydrophilic, and the water droplet will stick at the surface. If the contact angle approaches = 150° or more the surface is termed an ultra-hydrophobic/superhydrophobic surface. Similarly, as the contact angle approaches = 0° the water completely wets the surface, then the surface is termed a super-hydrophilic surface. The experimental results showed that TPE has a better self-cleaning ability than the other two materials. Changing the print layer height and extruder width produced samples that spanned relatively RQ ranges between (0.04–0.65 μm), yet there were significant differences in the wetting ability. For (TPU 98A), increasing layer height could reduce the wettability. It was shown that 0.10 mm of layer height had the lowest wettability (116°), whereas the maximum angle accrued at 0.15 mm of layer height (136°). This indicated that layer height changes caused an increase in wettability. The increase was considerable for one layer height that had been affected by the other parameters. The average value of the increase remained the same. For (TPC 45flex), the result showed that the wettability of 0.10 mm of layer height had the lowest value of the three materials (112°), and the maximum wettability was at 0.15 mm of layer height (143°). However, the (TPE) showed the highest values of 0.15 mm of layer height (149°), and the wettability for 0.10 mm of layer height (135°) was the lowest. Therefore, due to the connection between the roughness and wettability, the highest roughness values influenced the wettability, causing a higher contact angle for the three materials. The results are represented in the diagram in Figure 20.

Increasing the extruder width for (TPU 98A) increased the wettability values as 0.10 mm of layer height had the lowest wettability (116°), whereas the maximum angle accrued at 0.15 mm of layer height (136°). The increase was considerable for one extruder width, which had been affected by the other parameters. The average value of the increase remained the same. For (TPC 45flex), the result showed that the wettability of 0.10 mm of layer height had the lowest value of the three materials (112°), and the maximum wettability was at 0.15 mm of layer height (143°). However, the (TPE) showed the highest values of 0.15 mm of layer height (149°), and the wettability for 0.10 mm of layer height (135°) was the lowest, as shown in the diagram Figure 21.

There is a change for the same material with the two different orientation angles (45–90). Between 45° and 90°, the wettability changes according to the surface roughness changes. The three materials (TPU 98A), (TPC 45flex), and (TPE) showed that the highest wettability values were in a printing orientation of 90°, and the values decreased at a printing orientation of 45°, as shown in Figure 22.

##### Water Absorption and Fabric Wettability

The fabrication of the porous surface required a combination of the printing parameters. The pore size can influence the liquids in porous textiles behaviours and control the flow pattern of that liquid moving through a porous material. In this study, the flow of the water droplet through the printed textiles is caused by the surface movement, as the person wearing the fabric will be moving most of the time. This should provide the floating action as soon as the droplet comes to the surface. If the droplet is maintained on the fabric surface, it will absorb with time.

There is an absorption principle, but we believe that, having a self-cleaning wettability angle and the surface position keeps changing, there will be no time for absorption. Wettability will be a crucial parameter to describe how it is supposed to be self-cleaned. Additional experiments have been performed to determine the behaviour of wettability with time as shown in the figures below, Figure 23, Figure 24 and Figure 25. The contact angle of three materials without pores.

(TPU 98A).

**Figure 23 polymers-14-03128-f023:**
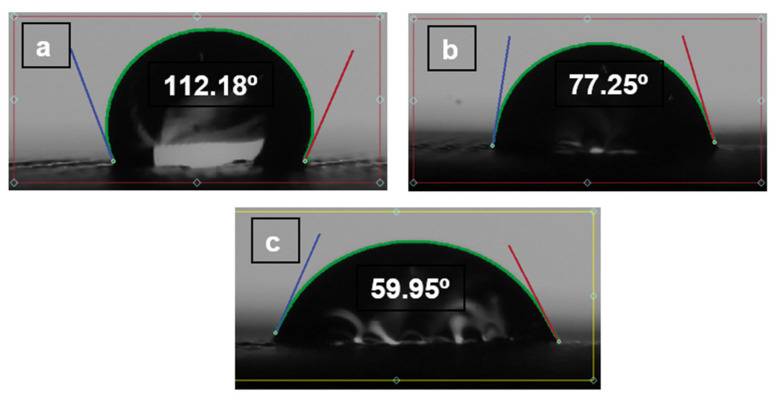
(**a**) The contact angle on the TPU. (**b**) The contact angle after 5 min. (**c**) The contact angle on after 10 min.

(TPE filaflex).

**Figure 24 polymers-14-03128-f024:**
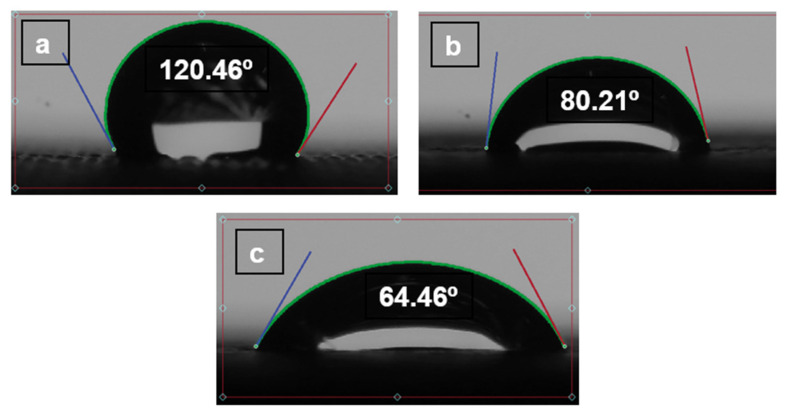
(**a**) The contact angle on the TPE. (**b**) The contact angle after 5 min. (**c**) The contact angle after 10 min.

(TPC 45flex).

**Figure 25 polymers-14-03128-f025:**
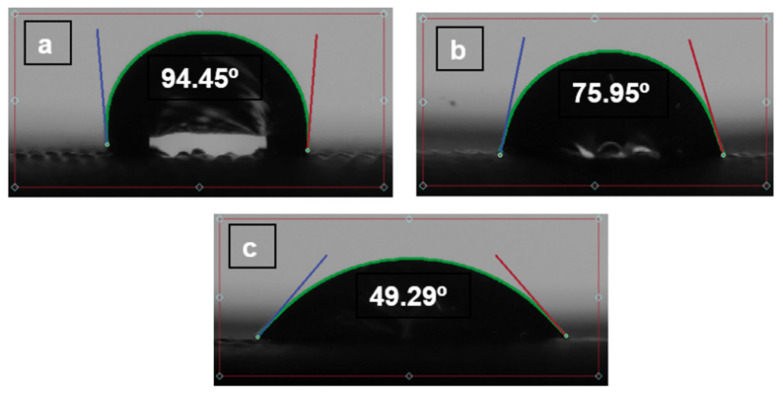
(**a**) The contact angle on the TPC. (**b**) The contact angle after 5 min. (**c**) The contact angle on after 10 min.

### 4.2. The Linear Regression for Different Printing Parameters

The microscopic features (porosity, roughness, and wettability) values were determined for different printing parameters experimentally along with the three materials: Thermoplastic Polyurethane (TPU 98A), Thermoplastic elastomer (TPE Filaflex), Thermoplastic Co-Polyester (TPC flex45). The regression equations were fitted by MATLAB.

(TPU 98A).


(4)
$$Porosity=48.12+0.03x_{1}+44.03x_{2}+8.19x_{3}\;$$



(5)
$$Roughness=-0.23+0.001x_{1}+1.20x_{2}+0.62x_{3}\;$$



(6)
$$Wettability=100.51+0.17x_{1}+137.22x_{2}+3.306x_{3}\;$$


The regression measurements of the three parameters are the positive values for the three features. The porosity (0.02), roughness (0.001) and wettability (0.172) are positive values. This proves that when building orientation changes from 45° to 90° the feature values increase. Similarly for the layer height and extruder regression measurements for porosity (44.03 and 8.19), roughness (1.20 and 0.63), and wettability (137.22, 3.30), which mean that the porosity and roughness values increase when the parameters also increase.

(TPE filaflex).


(7)
$$Porosity=3.14+0.03x_{1}+25.88x_{2}+13.06x_{3}\;$$



(8)
$$Roughness=-0.062+0.001x_{1}+1.29x_{2}+1.13x_{3}\;$$



(9)
$$Wettability=39.15+0.098x_{1}+139.11x_{2}+135.87x_{3}\;$$


The regression measurements of building orientation are positive values (0.027, 0.0014 and 0.098) for porosity, roughness, and wettability. This implies that, when building orientation changes from 45° to 90° it increases the feature values. The layer height regression coefficients are positive values (25.88, 1.29 and 139.11) for porosity, roughness and wettability which imply that increasing the parameter will increase the value. The regression coefficient of the extruder width is a positive value for roughness, porosity, and wettability (13.06, 1.13 and 135.87).

(TPC 45flex).


(10)
$$Porosity=53.17+0.03x_{1}+25.03x_{2}+8.25x_{3}\;$$



(11)
$$Roughness=-0.29+0.0009x_{1}+0.66x_{2}+0.81x_{3}\;$$



(12)
$$Wettability=93.02+0.11x_{1}+22.61x_{2}+45.21x_{3}\;$$


The regression measurements of building orientation are positive values (0.017 and 0.001) for porosity and roughness. This implies that, when building orientation increases from 45° to 90° the feature values increase. Similarly, the layer height regression coefficients are positive values (25.033 and 3.51) for porosity and roughness. The regression coefficient of the extruder width is a positive value only for surface roughness (8.24 and 1.65) it indicates that increasing the extruder width increases the value.

### 4.3. The Self-Cleaning Number

The self-cleaning number was evaluated based on the linear Equation (3).

Based on this equation if the wettability value =90° the total will be = 0. This indicates that the textile does not have self-cleaning ability and the water droplet will be absorbed by the surface. If the wettability value =150°, the total will be = 100, which means that the surface has the best self-cleaning ability, and the water droplet will roll off the surface. The scale will be from 0–95 as 0 means the textile fabric is not self-cleaned and 95 is the most optimal self-cloning ability.

(TPU 98A) showed that several scale values varied between 44.11, which is the lowest value with wettability of (0.10 mm LH, 0.3 mm EW and 45° O), and 73.49, which is the highest value of (0.15 mm LH, 0.5 mm EW and 90° O) for a self-cleaning number.(TPE filaflex) showed that several scale values varied between 55.22, which is the lowest value with wettability of (0.10 mm LH, 0.3 mm EW and 45° O), and 94.90, which is the highest value of (0.15 mm LH, 0.5 mm EW and 90° O) for a self-cleaning number.(TPC 45flex) showed that several scale values varied between 34.91, which is the lowest value with wettability of (0.10 mm LH, 0.3 mm EW and 45° O), and 72.33 which is the highest value of (0.15 mm LH, 0.5 mm EW and 90° O) for a self-cleaning number.

Comparing between the three materials the (TPE filaflex) showed the best self-cleaning result that provided the highest value of 94.90 with the combination of wettability (0.15 mm LH, 0.5 mm EW and 90° O).

### 4.4. The Optimization of Printing Parameters

The optimization of printing parameters was modelled to achieve the best self-cleaning ability of the printed specimens. The devised method and optimization model can be used to estimate the self-cleaning ability of printed fabric if the inputs of the printing parameters are known. It was found that we can attain even better self-cleaning behaviour if we provide various combinations of these parameters and establish high wettability with low porosity and high roughness hence a high self-cleaning number. The possible combinations of the printing parameters for optimal self-cleaning ability of all the three materials are discussed below.

TPU 98A

Equations (4) – (6) were used to find out the optimal self-cleaning ability for the TPU fabrics. Printing parameters were plotted with self-cleaning attributes as shown in Figure 26, Figure 27 and Figure 28. The minimum possible porosity was available on any value of orientation and lower values of EW (i.e., 0 to 0.2 mm) and LH (i.e., 0.25 to 0.3 mm). Maximum roughness was available on any orientation with higher values of EW (i.e., 0.9 to 1 mm) and LH (i.e., 0.45 to 0.5 mm). Similar trends could be observed for wettability, for orientation and LH. Maximum wettability seems independent from EW, bearing in mind that any contact angle above than 150 degrees for wettability measurement was suitable for self-cleaning. So, the optimum self-cleaning ability can be determined by selecting a decent value of roughness with a minimum possible porosity percentage. Therefore, in the case of TPU, the optimum self-cleaning can be obtained at any given orientation but with a compromise on desirable roughness and porosity percentage. The plots of roughness and porosity suggest that a good compromise can be made if the fabric is printed with an EW from 0.4 to 0.5 mm at any given LH.

TPE filaflex

The Equations (7)–(9) were used to find out the optimal self-cleaning ability for the TPU fabrics. Printing parameters were plotted with self-cleaning attributes as shown in Figure 29, Figure 30 and Figure 31. The minimum possible porosity was available on any value of orientation and lower values of EW (i.e., 0 to 0.2 mm) and LH (i.e., 0.25 to 0.3 mm). Maximum roughness was available on any orientation with higher values of EW (i.e., 0.9 to 1 mm) and LH (i.e., 0.45 to 0.5 mm). A similar trend could be observed for wettability, for orientation and LH. Maximum wettability seems independent to EW, bearing in mind that any contact angle above 150 degrees for wettability measurement was suitable for self-cleaning. So, the optimum self-cleaning ability can be determined by selecting a decent roughness value with a minimum possible porosity percentage. Therefore, in the case of TPE, the optimum self-cleaning could be obtained at any given orientation but with a compromise on desirable roughness and porosity percentage. The plots of roughness and porosity suggest that if the fabric was printed with an EW from 0.4 to 0.5 mm at any given LH it could be a good compromise.

TPC 45flex

The Equations (10)–(12) were used to find out the optimal self-cleaning ability for the TPU fabrics. Printing parameters were plotted with self-cleaning attributes as shown in Figure 32, Figure 33 and Figure 34. The minimum possible porosity was available on any value of orientation and lower values of EW (i.e., 0 to 0.2 mm) and LH (i.e., 0.25 to 0.3 mm). Maximum roughness was available on any orientation with higher values of EW (i.e., 0.9 to 1 mm) and LH (i.e., 0.45 to 0.5 mm). A similar trend could be observed for wettability for orientation and LH. Maximum wettability seems independent to EW. Any contact angle above 150 degrees for wettability measurement is suitable for self-cleaning. So, the optimum self-cleaning ability could be determined by selecting a decent roughness value with a minimum possible porosity percentage. The plots of roughness and porosity suggest that the fabric can be made with reasonable compromise at EW from 0.4 to 0.5 mm at any given LH. Therefore, for TPC, the optimum self-cleaning can be obtained at any given orientation but compromised with desirable roughness and porosity percentage.

## 5. Validation

To validate the results, we printed two specimens for the three materials with different printing parameter that had not been tested in the experimental scheme as shown in the Table 6 below.

As we clarified that the wettability was the most important parameter to justify the self-cleaning number, we measured the contact angle to the water’s droplet on the textile fabric surface experimentally to evaluate the self-cleaning number as shown in Table 7.

For the three martials the porosity values experimentally and based on Equations (4), (7) and (10) as shown in Table 8.

For the three martials the surface roughness values experimentally and based on Equations (5), (8) and (11) as shown in Table 9.

The self-cleaning number evaluated based on the linear Equation (3) as shown in Table 10.

## 6. Conclusions

In the present work, an experimental study was performed on (FFF) using three different thermoplastic flexible filaments printing materials: thermoplastic polyurethane (TPU 98A), thermoplastic elastomers (TPE felaflex), and thermoplastic co-polyester (TPC flex45) to fabricate 3D printed polymeric textile fabrics.

It was found that the printing parameters have a very significant effect on the self-cleaning properties when optimizing the selection of the process parameter combination of layer height, extruder width, and printing orientation. The results showed that changing layer height and extruder width combined can affect the porosity, surface roughness, and wettability. The highest layer height has a rougher impact on the surface texture.The changes in porosity numbers are dependent on the material itself rather than the layer height changes. They show porosity values obtained between 9% and 18% for the chosen layer height. They slightly change by about 2% to 5% at higher and lower layer height, respectively.The lowest surface roughness occurs on 0.10 mm of layer height for TPC with (112°) wettability contact angle, whereas the highest occurs on the 0.15 mm of layer height for TPE with (149°) wettability contact angle. Changing the print layer height and extruder width produces samples that span relative roughness ranges between (0.04–0.65 μm).The experimental results showed that TPE has a better self-cleaning ability than the other two materials. It showed that several scale values varied between 55.22, which was the lowest value with wettability of (0.10 mm LH, 0.3 mm EW and 45° O), and 94.90, which was the highest value of (0.15 mm LH, 0.5 mm EW and 90° O) for self-cleaning number.Printing different layer heights can affect the printing time and divide a 3D model into more layers affecting the quality of the printed fabric. Extruder width was found to be much more critical for surface printed quality, as lowering the width deteriorated the printed surface, causing more threads to be printed to cover the surface. A desirable structural geometry and fiber orientation are achievable with reasonable and accurate control of printing parameters.

The future of polymeric self-cleaning textile with its aesthetic look can be used and fabricated by controlling the printing parameters as the discussed model describes. The devised method and model can be used to estimate the self-cleaning ability of printed fabric if the inputs of the printing parameters are known.

## Figures and Tables

**Figure 1 polymers-14-03128-f001:**
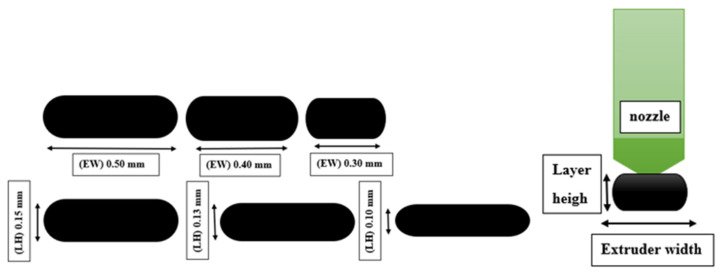
Layer height and extruder width definition.

**Figure 2 polymers-14-03128-f002:**
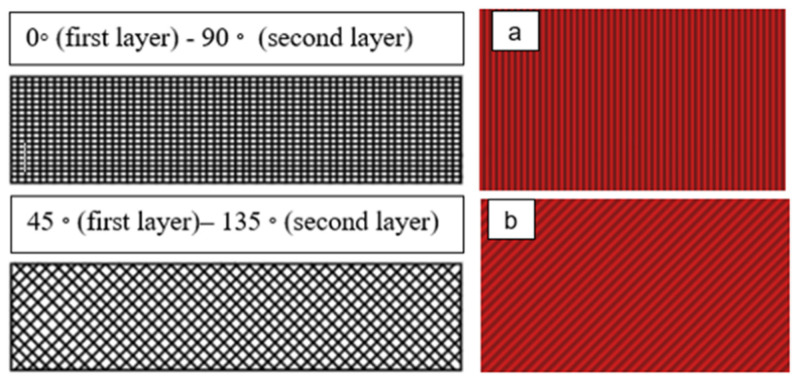
Scheme showing two samples printed flat and loaded in the x/y direction 45 degree–90 degree. (**a**) XY (45°) and (**b**) XY (90°).

**Figure 3 polymers-14-03128-f003:**
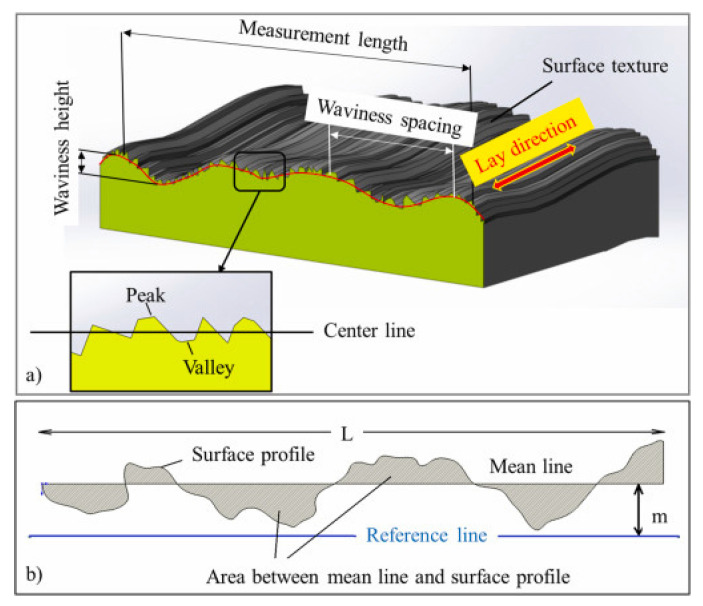
Terminology illustration of (**a**) surface texture (**b**) surface roughness (Ra) [36].

**Figure 4 polymers-14-03128-f004:**
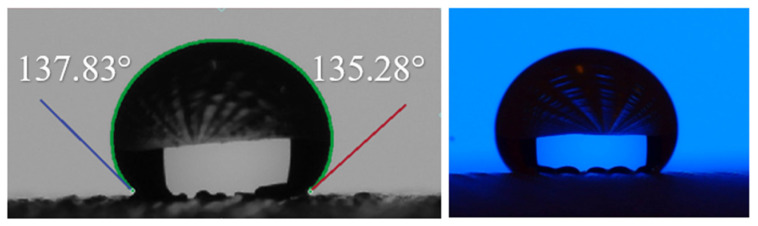
The image and measurement of contact angle of a water droplet on fabric surface.

**Figure 5 polymers-14-03128-f005:**
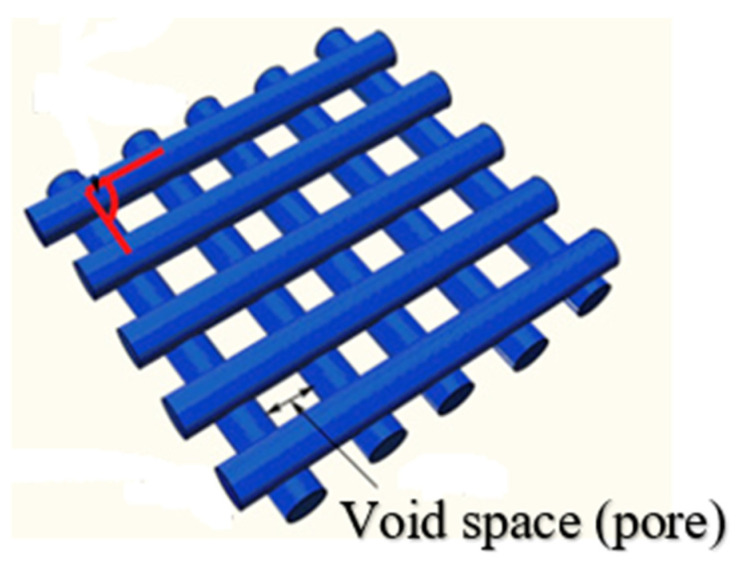
The porosity of textile fabrics.

**Figure 6 polymers-14-03128-f006:**
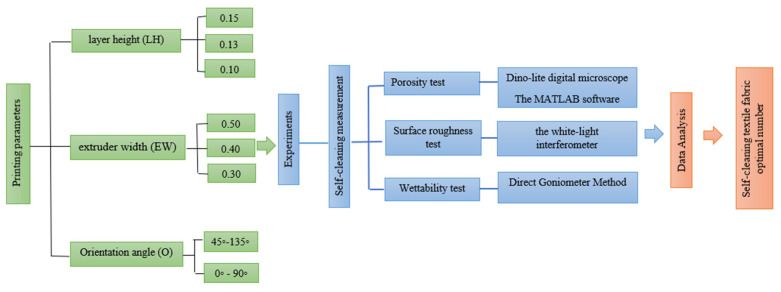
The design of the experiments.

**Figure 7 polymers-14-03128-f007:**
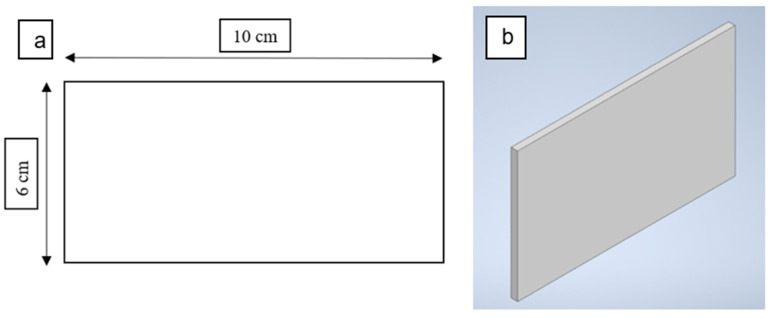
(**a**) The dimension of specimen test (**b**) The CAD model of the specimen.

**Figure 8 polymers-14-03128-f008:**
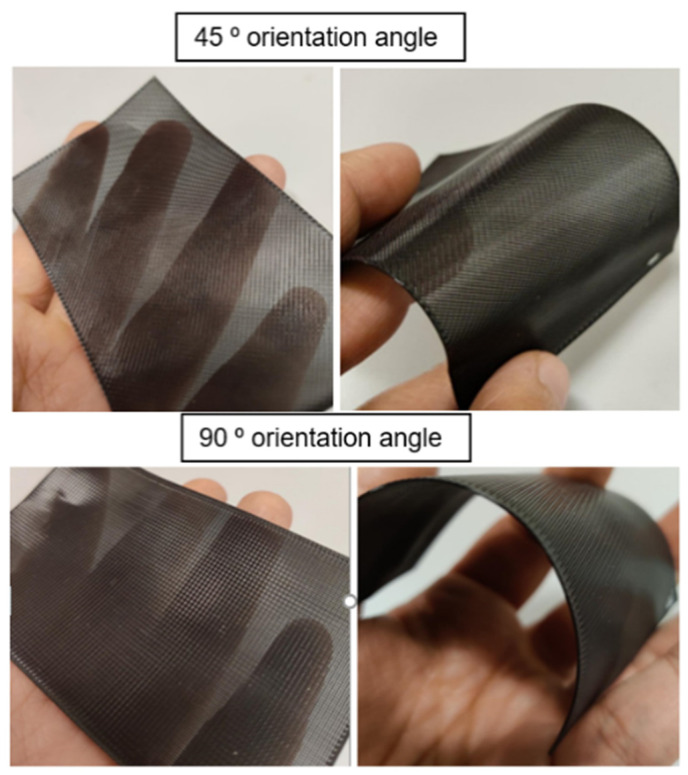
The self-cleaning textile with both orientation angles 45–90°.

**Figure 9 polymers-14-03128-f009:**
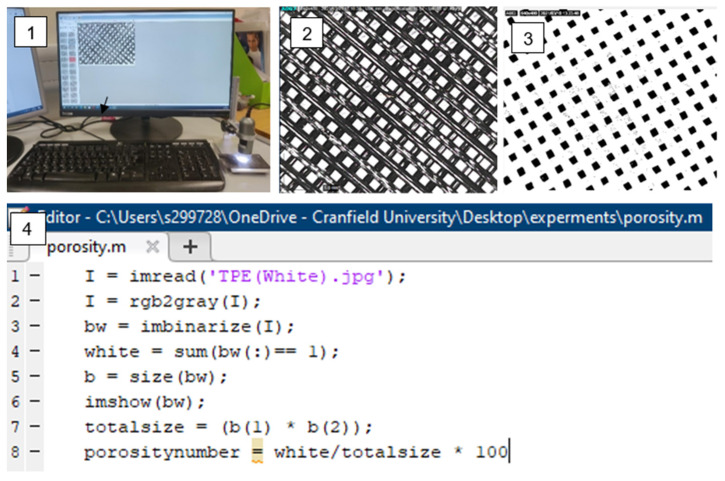
Porosity measurement method.

**Figure 10 polymers-14-03128-f010:**
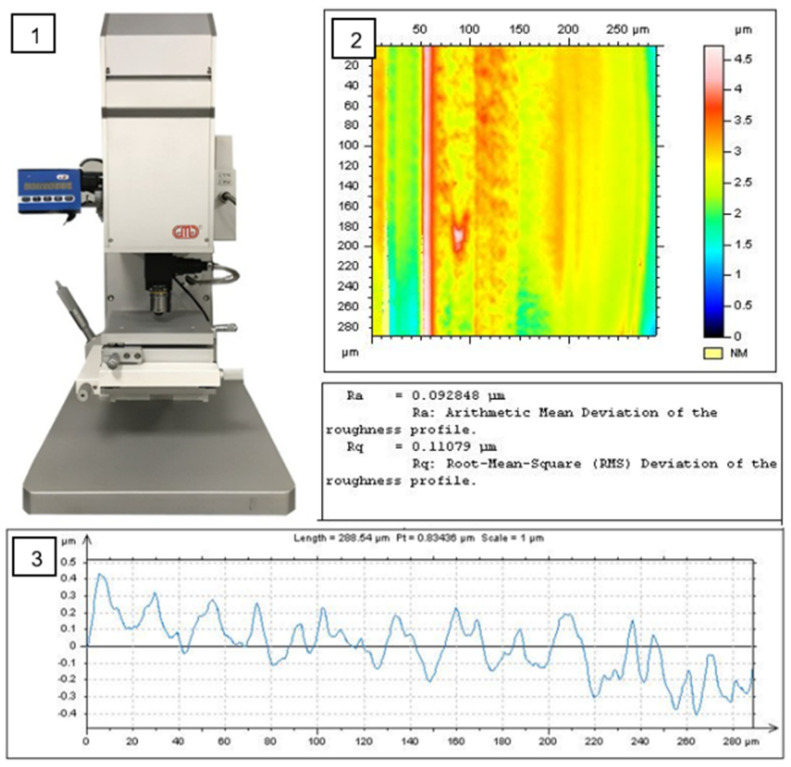
Surface roughness measurement method.

**Figure 11 polymers-14-03128-f011:**
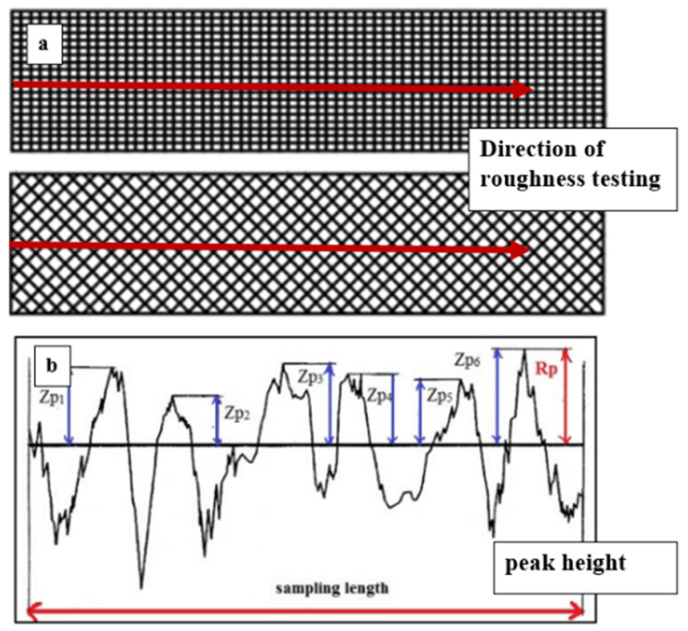
(**a**) The measurement direction of roughness testing (**b**) The peak height profile.

**Figure 12 polymers-14-03128-f012:**
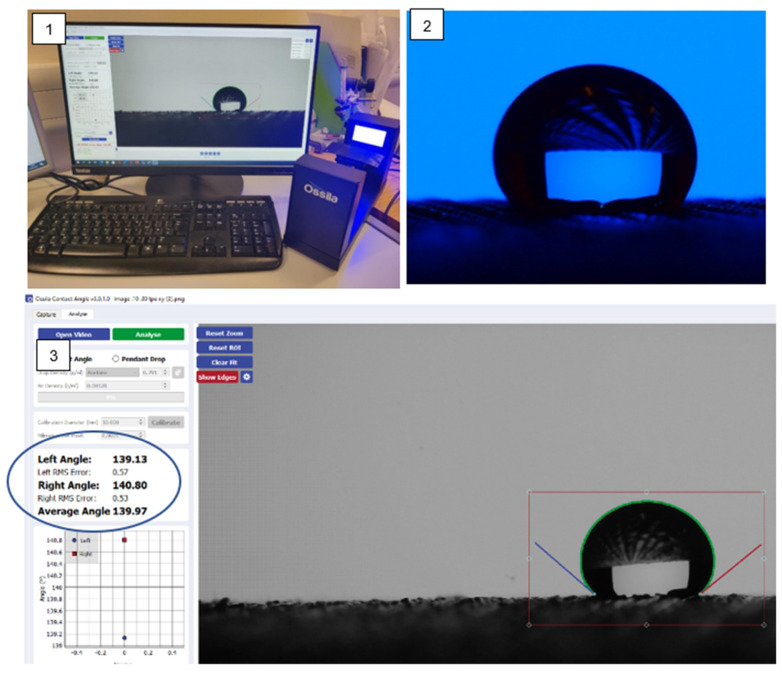
Wettability measurement method.

**Figure 13 polymers-14-03128-f013:**
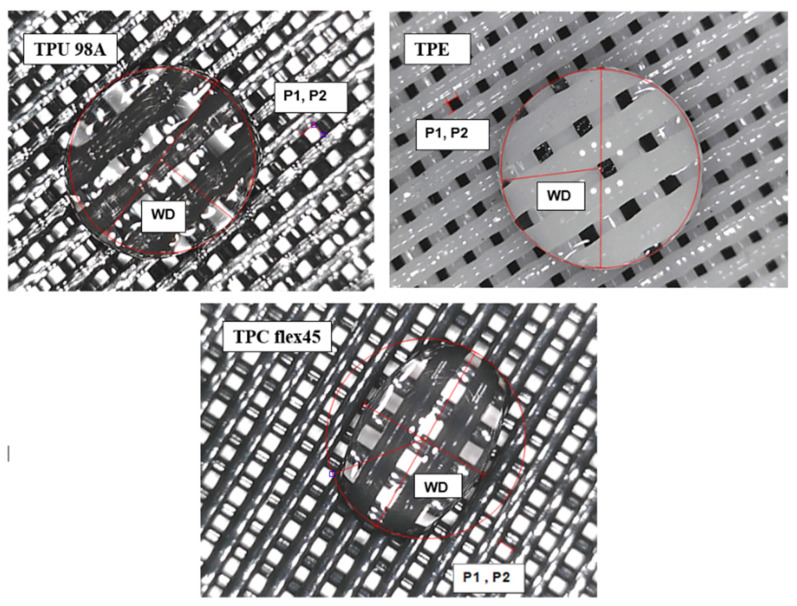
The Unit Cell measure.

**Figure 14 polymers-14-03128-f014:**
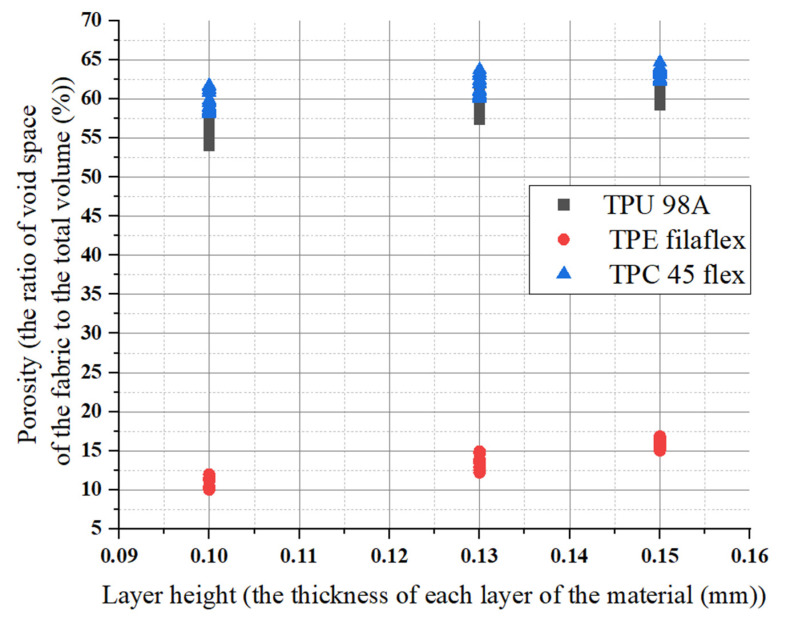
Layer height influences on porosity for the three materials.

**Figure 15 polymers-14-03128-f015:**
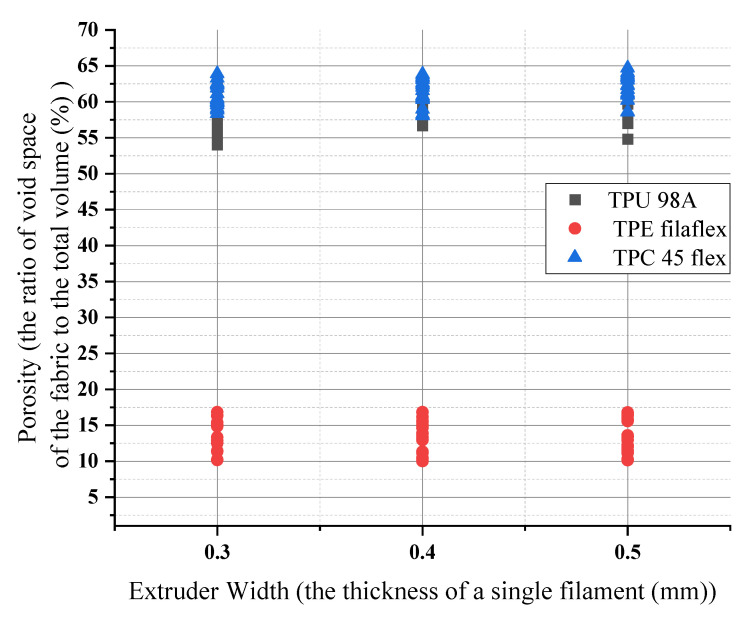
Extruder width influences on porosity numbers for the three materials.

**Figure 16 polymers-14-03128-f016:**
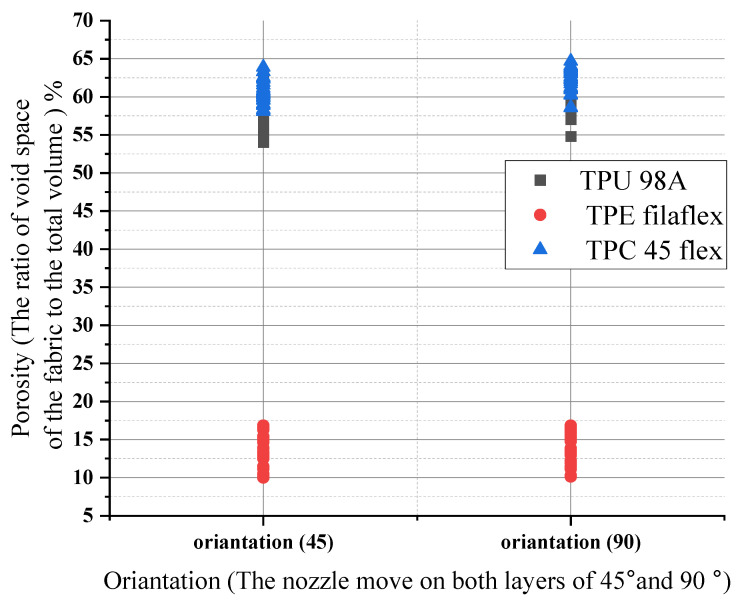
Orientation influences on porosity numbers for the three materials.

**Figure 17 polymers-14-03128-f017:**
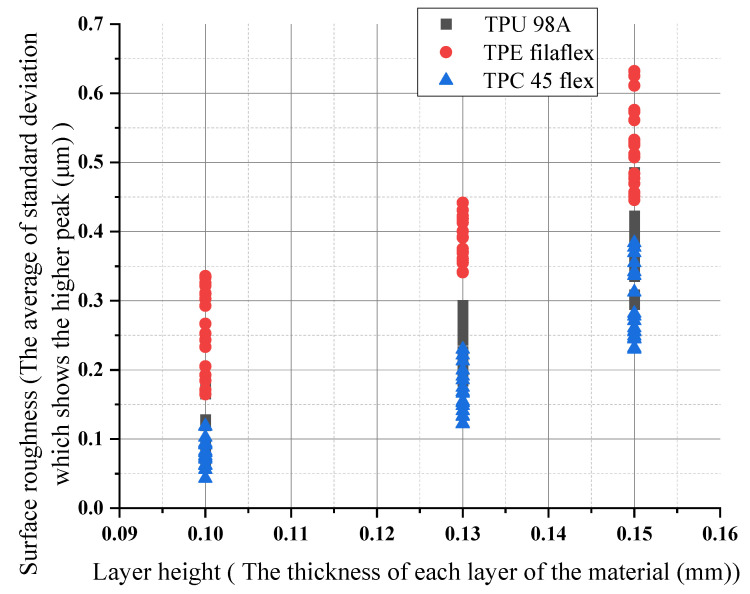
Layer height influences on surface roughness for the three materials.

**Figure 18 polymers-14-03128-f018:**
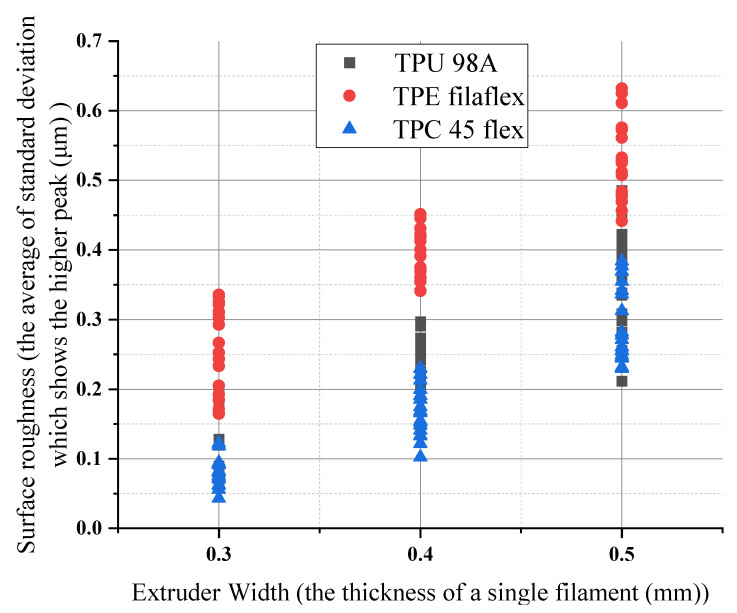
Extruder width influences on surface roughness for the three materials.

**Figure 19 polymers-14-03128-f019:**
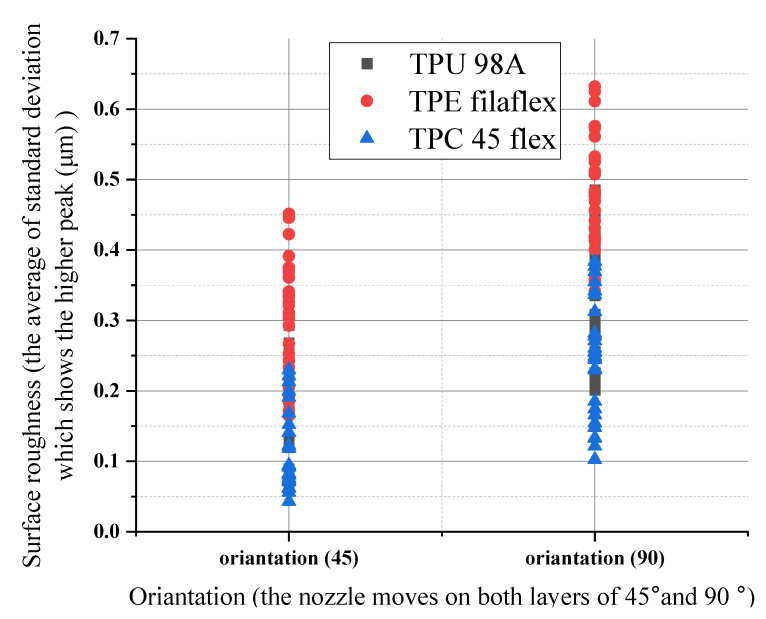
Orientation influences on surface roughness for the three materials.

**Figure 20 polymers-14-03128-f020:**
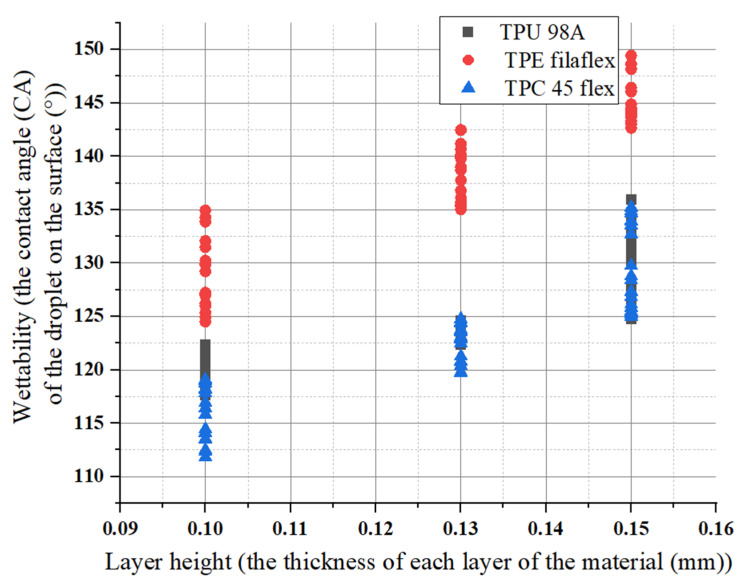
Layer height influences on wettability for the three materials.

**Figure 21 polymers-14-03128-f021:**
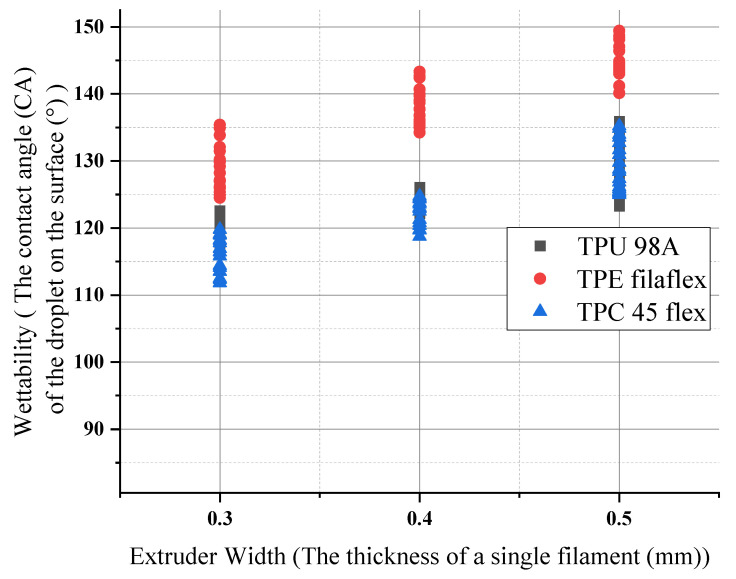
Extruder width influences on wettability for the three materials.

**Figure 22 polymers-14-03128-f022:**
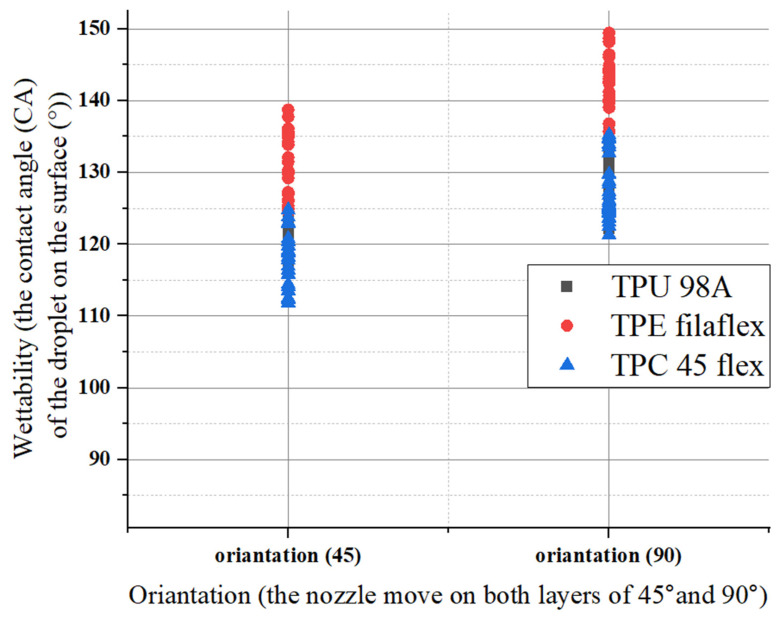
Orientation influences on wettability for the three materials.

**Figure 26 polymers-14-03128-f026:**
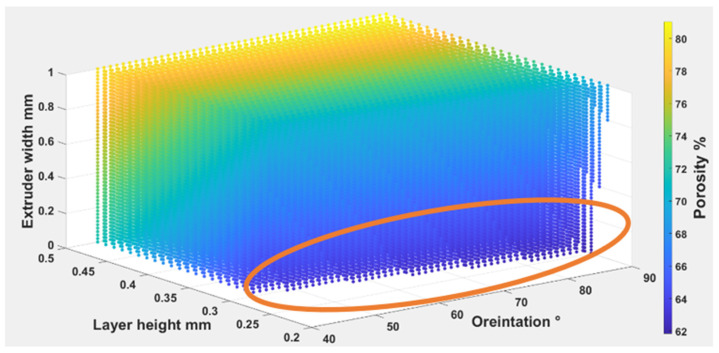
The possible combination of the printing parameters range for (O), (LH), (EW) to achieve the best self-cleaning performance for porosity.

**Figure 27 polymers-14-03128-f027:**
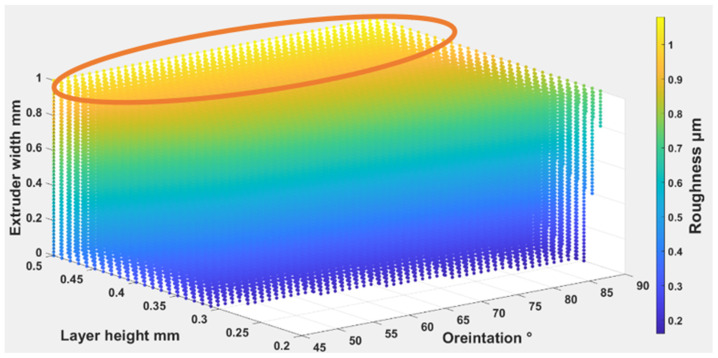
The possible combination of the printing parameters range for (O), (LH), (EW) to achieve the best self-cleaning performance for roughness.

**Figure 28 polymers-14-03128-f028:**
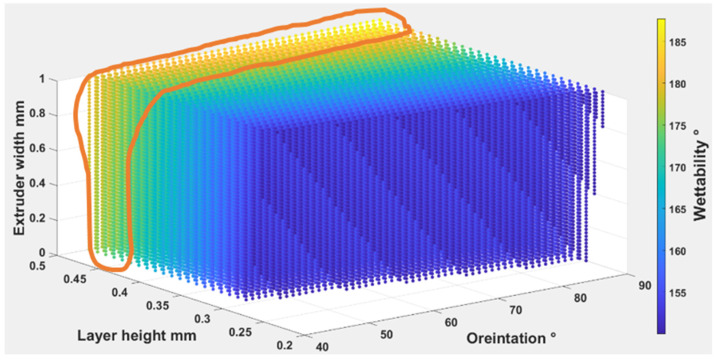
The possible combination of the printing parameters range for (O), (LH), (EW) to achieve the best self-cleaning performance for wettability.

**Figure 29 polymers-14-03128-f029:**
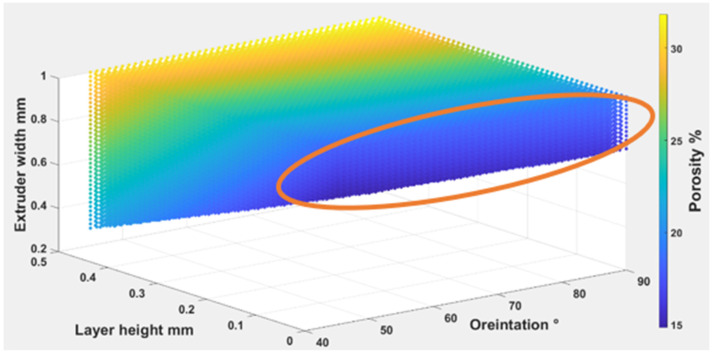
The possible combination of the printing parameters range for (O), (LH), (EW) to achieve the best self-cleaning performance for porosity.

**Figure 30 polymers-14-03128-f030:**
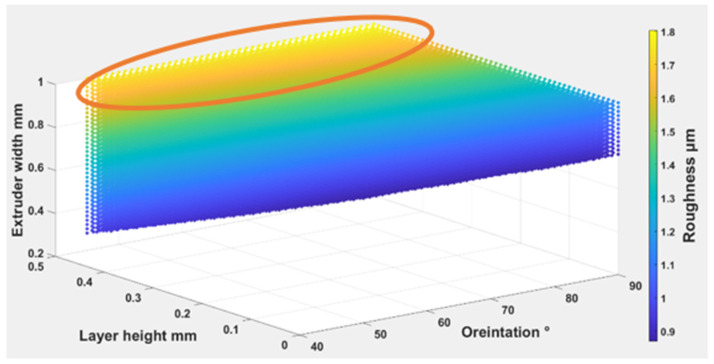
The possible combination of the printing parameters range for (O), (LH), (EW) to achieve the best self-cleaning performance for roughness.

**Figure 31 polymers-14-03128-f031:**
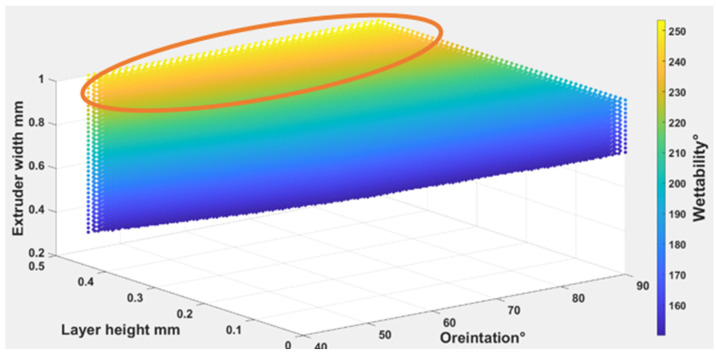
The possible combination of the printing parameters range for (O), (LH), (EW) to achieve the best self-cleaning performance for wettability.

**Figure 32 polymers-14-03128-f032:**
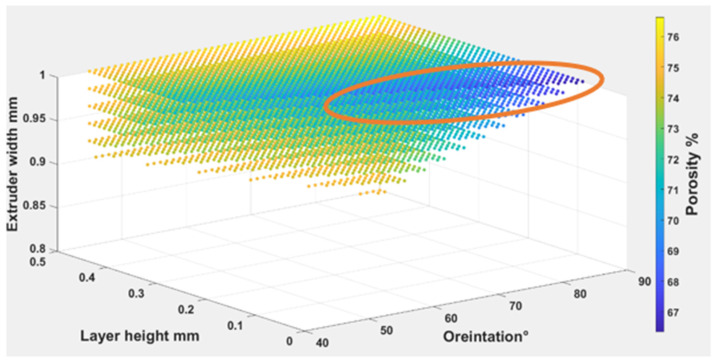
The possible combination of the printing parameters range for (O), (LH), (EW) to achieve the best self-cleaning performance for porosity.

**Figure 33 polymers-14-03128-f033:**
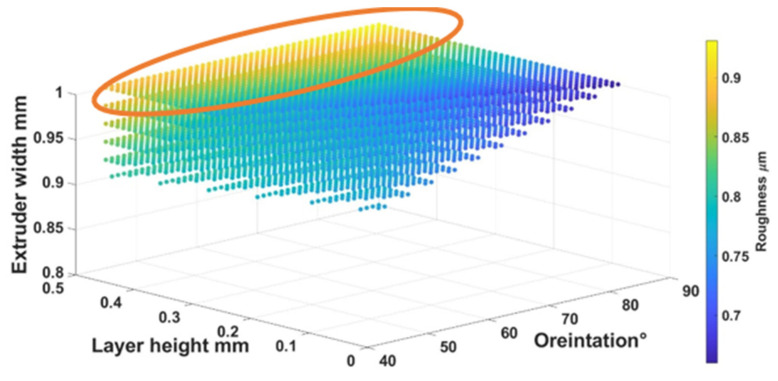
The possible combination of the printing parameters range for (O), (LH), (EW) to achieve the best self-cleaning performance for roughness.

**Figure 34 polymers-14-03128-f034:**
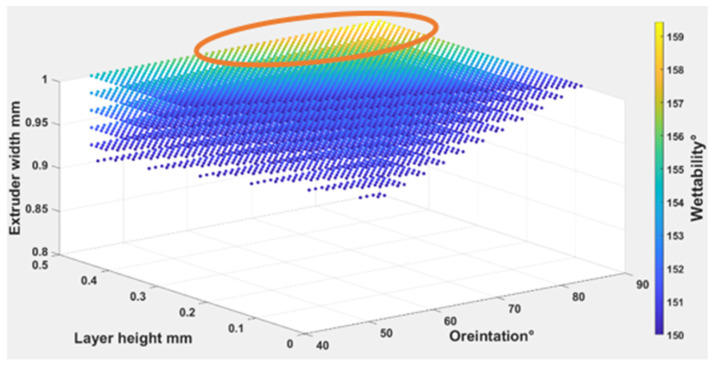
The possible combination of the printing parameters range for (O), (LH), (EW) to achieve the best self-cleaning performance for wettability.

**Table 1 polymers-14-03128-t001:** Printing parameters.

Layer Height (LH) mm	Extruder Width (EW) mm	Orientation Angle (O) Degree °
0.15	0.50	0° (first layer)–90° (second layer), 45° (first layer)–135° (second layer)
0.13	0.40	0–90°, 45–135°
0.10	0.30	0–90°, 45–135°

**Table 2 polymers-14-03128-t002:** The flexible filaments specifications and properties of the (FFF)-based.

Properties/Filament	TPU 98A	TPE Fila Flex	TPC Flex 45
Chemical name/blend	Thermoplastic polyurethane	Thermo Plastic Elastomer	Thermoplastic Co-Polyester
Cost per kilo(£/kg)	22–40	20–30	30–60
supplier	RS Components	RS Components	RS Components
Diameter (mm)	1.75	1.75	1.75
Printing temperature (°C)	205–230	210–240	210–250
Density (g/m^3^)	1.14	1.09	1.16

**Table 3 polymers-14-03128-t003:** The 3D parameters were used to produce the samples.

Filament	Layer Height (mm)	Number of Layers	Extruder Width (mm)	Orientation Degree (°)	Number of Samples
TPU 98A	0.15	2	0.50	45–135° 0–90°	3
			0.40		
			0.30		
	0.13	2	0.50	45–135° 0–90°	3
			0.40		
			0.30		
	0.10	2	0.50	45–135° 0–90°	3
			0.40		
			0.30		
TPE Filaflex	0.15	2	0.50	45–135° 0–90°	3
			0.40		
			0.30		
	0.13	2	0.50	45–135° 0–90°	3
			0.40		
			0.30		
	0.10	2	0.50	45–135° 0–90°	3
			0.40		
			0.30		
TPC 45 Flex	0.15	2	0.50	45–135° 0–90°	3
			0.40		
			0.30		
	0.13	2	0.50	45–135° 0–90°	3
			0.40		
			0.30		
	0.10	2	0.50	45–135° 0–90°	3
			0.40		
			0.30		

**Table 4 polymers-14-03128-t004:** The Unit Cell measurements.

Name	TPU 98A	TPE fila flex	TPC 45 flex
Porosity (P1) length (mm)	0.044	0.035	0.047
Porosity (P2) width (mm)	0.044	0.035	0.047
Water Droplet (WD)	Circumference (mm)	Circumference (mm)	Circumference (mm)
	1.867	1.928	1.932
	Area (mm)	Area (mm)	Area (mm)
	0.277	0.296	0.297
	Radius (mm)	Radius (mm)	Radius (mm)
	0.297	0.307	0.307
	Diameter (mm)	Diameter (mm)	Diameter (mm)
	0.594	0.614	0.615

**Table 5 polymers-14-03128-t005:** Standardized Value for different parameters.

Layer Height (LH) mm	Extruder Width (EW) mm	Orientation Angle (O)°	Standardized Value
0.10	0.30	-	−1
0.13	0.40	45°	0
0.15	0.50	90°	1

**Table 6 polymers-14-03128-t006:** The 3D parameters were used to produce the samples.

Layer Height (LH) mm	Extruder Width (EW) mm	Orientation Angle (O) Degree °
0.14	0.60	70° (first layer)–210° (second layer)

**Table 7 polymers-14-03128-t007:** The wettability measurement for the samples.

**(TPU)—Orientation 70°**					
**Sample**	**(LH)** **mm**	**(EW)** **mm**	**Wettability** **Experimentally** **Degree °**	**Wettability** **Based on Equation (6)** **Degree °**	**R-Square** **(** **%** **)**
1	0.14	0.6	133.52	133.7301	−0.1573
2	0.14	0.6	132.34	133.7301	−1.0504
**(TPE)** **—** **Orientation 70°**					
**Sample**	**(LH)** **mm**	**(EW)** **mm**	**Wettability** **Experimentally** **Degree °**	**Wettability** **Based on Equation (9)** **Degree °**	**R-Square** **(** **%** **)**
1	0.14	0.6	146.79	147.0002	−0.1432
2	0.14	0.6	146.8	147.0002	−0.1364
**(TPC)—Orientation 70°**					
**Sample**	**(LH)** **mm**	**(EW)** **mm**	**Wettability** **Experimentally** **Degree °**	**Wettability** **Based on Equation (12)** **Degree °**	**R-Square** **(** **%** **)**
1	0.14	0.6	131.52	130.8233	0.5297
2	0.14	0.6	130.34	130.8233	−0.3708

**Table 8 polymers-14-03128-t008:** The porosity measurement for the samples.

Samples	(O) Degree °	(LH) mm	(EW) mm	Porosity Experiment (%)	Porosity Based on the Equations (%)	R-Square (%)
TPU 1	70	0.14	0.6	60.5487	61.18675	−1.0538
TPU 2	70	0.14	0.6	61.3495	61.18675	0.2653
TPE 1	70	0.14	0.6	15.9889	16.5023	0.8902
TPE 2	70	0.14	0.6	16.2713	16.5023	0.9843
TPC 1	70	0.14	0.6	62.2025	63.46319	−2.0267
TPC 2	70	0.14	0.6	61.1086	63.46319	−3.8531

**Table 9 polymers-14-03128-t009:** The surface roughness measurement for the samples.

Samples	(O) Degree °	(LH) mm	(EW) mm	Surface Roughness Experiment μm	Surface Roughness Based on the Equations μm	R-Square (%)
TPU 1	70	0.14	0.6	0.37949	0.383772	−1.1284
TPU 2	70	0.14	0.6	0.38049	0.383772	−0.8626
TPE 1	70	0.14	0.6	0.63273	0.64494	−1.9297
TPE 2	70	0.14	0.6	0.64253	0.64494	−0.3751
TPC 1	70	0.14	0.6	0.35156	0.357016	−1.5519
TPC 2	70	0.14	0.6	0.34996	0.357016	−2.0162

**Table 10 polymers-14-03128-t010:** The self-cleaning number for the samples.

Samples	(O) Degree °	(LH) mm	(EW) mm	Wettability Degree °	Self-Cleaning Number
TPU 1	70	0.14	0.6	133.52	69.632
TPU 2	70	0.14	0.6	132.34	67.744
TPE 1	70	0.14	0.6	146.79	90.864
TPE 2	70	0.14	0.6	146.8	90.88
TPC 1	70	0.14	0.6	131.52	66.432
TPC 2	70	0.14	0.6	130.34	64.544

## Data Availability

The data presented in this study are available on request from the corresponding authors.

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
