# Peer review of "Influence of Printing Parameters on Self-Cleaning Properties of 3D Printed Polymeric Fabrics"

_polymers, 2022, doi:10.3390/polym14153128_

Round 1

Reviewer 1 Report

1.The authors should concern on the references - Pg 4/16

2. Abstract has to be more precise

3. The nuber of variables chosen has to be increased in order to a come to a fair conclusion

4. Trends when varying the parameters have to be clearly understood and presented

5. Conclusion has to be more precise , accurate and quantitative.

Reviewer 2 Report

This manuscript reports the impact of printing parameters on the surface properties of textile fabrics. Interesting results are presented.

The manuscript will need revision for accuracy, clarity and readability. Corrections are penciled-in directly on pages of the manuscript attached. These are illustrative of the kinds of changes needed throughout. In rewriting, careful attention should be paid to the use of articles, tenses and proper sentence structure. Often what is intended is not stated. For example, printed textile fabrics were not fabricated "in this paper" but rather in the laboratory. Results are presented in the paper. Similarly, "most textile fabrics use surface coating methods" should be "surface coating methods were used ..." Textile fabrics are not active and lack the ability to "use." Often "by" should be "using." These kinds of things need to be corrected throughout. The manuscript will strongly benefit from careful editing.

Reviewer 3 Report

This manuscript demonstrates the potential of controlling self-cleaning capability of fabrics by adjusting three 3D printing parameters: the layer height, extruder width and fiber orientation. Authors adjust these three parameters and try to find the pattern of change on self-cleaning behavior with three different polymer materials. This is a good attempt, however, there are major issues with this work, and they need to be addressed prior to publication.

Major issue 1

What is the contact angle of three materials individually without pores? I could not find any data of these three materials listed in the manuscript. The reason why I ask this question is that the wettability of porous materials is largely dependent on the hydrophilicity of raw material.

The 19th citation in this manuscript talks about SLIPS, which is a liquid infused porous materials such that contact angle of a droplet on such surface will change along with time. Authors should refer to origin of SLIPS technique:

Wong, Tak-Sing, et al. "Bioinspired self-repairing slippery surfaces with pressure-stable omniphobicity." Nature 477.7365 (2011): 443-447.

This same principle can be applied to this work. If the raw material, TPU 98A, TPE filaflex or TPC 45flex, is hydrophilic, the printed porous material will automatically absorb water along with time to generate a liquid infused layer and the contact angle of the droplet will diminish after some time. To validate such phenomenon does not occur, authors should conduct additional experiments to demonstrate the stability of contact angles on porous materials made from these three materials.

Major issue 2

I fail to understand why porosity changed by layer height has influence on contact angle. Contact angle is a surface property directly related to the interaction among the material surface, liquid and air, which is defined by Young’s equation. The bulk property does not have anything to do with the contact angle. Therefore, the porosity defined in by authors as Y = b0 + b1x1 + b2x2 + b3x3 doesn’t seem to be the correct parameter to influence the contact angle where b1x1 here should not have any impact. The only case when b1x1 has an impact is when hydrophilicity plays a role where water on the surface penetrate inside the bulk. However, this is the case described in Major issue 1.

Overall, these two major issues make the manuscript scientific wrong in a way that the data were statistically analyzed prior to taking account of additional scientific phenomenon while fail to exclude unrelated factor. Authors should conduct additional experiments and re-analyze the data using a new model.

Minor issues

1.     Why choose these three parameters? There’s only citations listed in the introductions without analyzing. Authors should illustrate why they choose these 3 parameters instead of just saying because other people choose them, we choose them as well.

2.     There are some references not shown in the manuscript. I believe authors use some software to insert citations while some citations are not shown. Instead, it’s shown as ‘Error! Reference source not found’.

Round 2

Reviewer 2 Report

This manuscript is improved.